# The Wildland Fire Heat Budget—Using Bi-Directional Probes to Measure Sensible Heat Flux and Energy in Surface Fires

**DOI:** 10.3390/s21062135

**Published:** 2021-03-18

**Authors:** Matthew B. Dickinson, Cyle E. Wold, Bret W. Butler, Robert L. Kremens, Daniel Jimenez, Paul Sopko, Joseph J. O’Brien

**Affiliations:** 1Forestry Sciences Lab, USDA Forest Service, Northern Research Station, 359 Main Road, Delaware, OH 43015, USA; 2Missoula Fire Sciences Laboratory, USDA Forest Service, Rocky Mountain Research Station, 5775 US Highway 10 W, Missoula, MT 59808, USA; cyle.wold@usda.gov (C.E.W.); bret.butler@usda.gov (B.W.B.); dan.jimenez@usda.gov (D.J.); paul.sopko@usda.gov (P.S.); 3Chester F. Carlson Center for Imaging Science, Rochester Institute of Technology, One Lomb Memorial Drive, Rochester, NY 14623, USA; kremens@cis.rit.edu; 4Center for Disturbance Science, USDA Forest Service, Southern Research Station, 320 Green Street, Athens, GA 30602, USA; joseph.j.obrien@usda.gov

**Keywords:** wildland fire, fire energy, sensible heat flux, convective heat flux, fire radiated energy (FRE), residence time, bi-directional probe, flame velocity, gas temperature, RxCADRE Project

## Abstract

Sensible energy is the primary mode of heat dissipation from combustion in wildland surface fires. However, despite its importance to fire dynamics, smoke transport, and in determining ecological effects, it is not routinely measured. McCaffrey and Heskestad (A robust bidirectional low-velocity probe for flame and fire application. Combustion and Flame 26:125–127, 1976) describe measurements of flame velocity from a bi-directional probe which, when combined with gas temperature measurements, can be used to estimate sensible heat fluxes. In this first field application of bi-directional probes, we describe vertical and horizontal sensible heat fluxes during the RxCADRE experimental surface fires in longleaf pine savanna and open ranges at Eglin Air Force Base, Florida. Flame-front sensible energy is the time-integral of heat flux over a residence time, here defined by the rise in gas temperatures above ambient. Horizontal flow velocities and energies were larger than vertical velocities and energies. Sensible heat flux and energy measurements were coordinated with overhead radiometer measurements from which we estimated fire energy (total energy generated by combustion) under the assumption that 17% of fire energy is radiated. In approximation, horizontal, vertical, and resultant sensible energies averaged 75%, 54%, and 64%, respectively, of fire energy. While promising, measurement challenges remain, including obtaining accurate gas and velocity measurements and capturing three-dimensional flow in the field.

## 1. Introduction

The wildland fire heat budget balances heat generated by combustion (which is reduced by inefficiencies), heat sinks associated with heating fuels to ignition, and heat dissipation [1]. Based on limited measurements, sensible heat flux (kW/m^2^) and energy (kJ/m^2^), its time-integral, have been found to dissipate more energy from wildland fires than other modes, including radiation [2,3], and we expect that radiated energy, the next largest contributor, will be limited to about 10–20% of fire energy (i.e., total energy generated by combustion [1,2,3,4,5]). Sensible heat flux is the transport of heat in hot gases across a reference plane. The term has roots in the meteorological literature (e.g., [6]) while it is often referred to as convective or advective flux in the engineering literature (e.g., [7,8]). Based on flow velocities and gas temperatures, sensible heat fluxes in wildland fires have been measured using videography [9,10], particle velocimetry [11,12], one-dimensional anemometry [13,14], and two- and three-dimensional anemometry [15,16,17]. Sensible heat flux and energy have not been estimated in or just above wildland fire flames because instrumentation has not been sufficiently fire-hardened [18,19]. Originating in building fire research, McCaffrey and Heskestad [20] describe an instrument for measuring flow velocities in flames based on pressure differentials between dynamic (facing on-coming flow) and static (sheltered) orifices. The instrument has the merit that it can be made resistant to the harsh environment of flames.

We can, in a coarse way, define sensible heat flux and energy by considering a control volume that encloses the fuel, flames, heated air and combustion gas products (plume) and soil heated by the fire. All the energy generation from combustion is contained in this volume. A heat budget for the control volume roughly balances heat sources, sinks, and dissipation integrated over the time period from ignition through the cool-down to ambient temperatures (see [1]):(1)$$Wh_{C}\left(1-\varphi \right)=WQ_{P}+E_{G}+E_{V}+E_{L}+E_{R}+E_{S}$$
where *W* is the areal fuel consumption on a moisture and ash-free basis (kg/m^2^); *h_C_* is the high heat of combustion (i.e., includes heat of condensation of water generated by the combustion process, kJ kg^–1^); φ is the fractional reduction in *h_C_* because of incomplete combustion, *h_C_*(1 − φ) is the effective heat of combustion [21,22]; *Q_P_* is the fuel pre-heating and pyrolysis enthalpy (kJ/kg), defined so as not to include the fuel moisture vaporization enthalpy, see below); *E_G_* is the areal energy density transferred into the soil (kJ/m^2^); *E_V_* is the energy transferred to vegetation (e.g., tree stems, branches, and foliage); *E_L_* is the fire latent energy density (kJ/m^2^), the condensation energy in water vapor generated from both fuel moisture and the combustion process; *E_R_* is the fire radiative energy density (kJ/m^2^), i.e., the radiated flux time-integrated over the period from ignition to cool-down; and *E_S_* is sensible energy density transported by the buoyancy-driven rise of heated combustion products and directly heated air and smoke. Both sides of Equation (1) have units of areal energy density, kJ/m^2^. The left side of Equation (1) is the energy available to do work near the fire front which we term ‘fire energy’. The heat sink and dissipation modes on the right-hand side are ordered by their expected relative magnitude in flame fronts, although we acknowledge that the ordering is partly conjecture ([1]). All quantities involving fuel mass (e.g., consumption, heats of combustion) are on an ash- and moisture-free basis.

The integrated heat budget is coarse in the sense that heat dissipation modes are not independent and their magnitudes are dependent on the control volume [1]. For instance, sensible energy is reduced by radiation from the hot flame and plume. Sensible energy is likely to be a larger proportion of fire energy if the control volume is occupied primarily by flame. Heat and momentum from plumes above surface fires is transferred to forest canopies [23] but in a control volume that primarily contains the flaming front, that transfer is ignored. Net heat flux at the soil surface will be determined by gas-phase conduction in the burning fuel bed, radiative and convective heat transfer to the soil surface (as mediated by burning fuels and ash), and heat losses by radiation and convection.

In this paper, we describe measurements of sensible heat flux and energy (Equation (1)) based on in situ instruments in wildland surface fires. Sensible heat fluxes and energy have a particular relevance to fire propagation [24], plume dynamics [6,17], and various ecological fire effects [25], including tree crown injury [26,27,28], faunal exposures [29,30], and airborne transport of microbes [31]. This study is a first application of bi-directional probes for measuring gas flow velocities in and just above wildland fire flames. We estimate that sensible energy accounted for more than 50% of fire energy (left-hand side of Equation (1)). We discuss measurement uncertainties and improvements that can be made in future experiments. Our paper is an important step towards closing the wildland fire heat budget [1] which provides a standard for assessing measurements and is expected to lead to advances in understanding and predicting fire dynamics, plume dynamics, and fire effects.

## 2. Materials and Methods

### 2.1. Study Site and Fire Behavior

Data were collected in early November 2012 within burn blocks in an 8-km x 4-km area of Eglin Air Force Base in northwestern Florida during the Prescribed Fire Combustion and Atmospheric Dynamics Research Experiment (RxCADRE), a coordinated measurements campaign described in Ottmar et al. [32] and associated papers. Burn blocks were characterized by either an herbaceous and shrub fuel mix maintained as open range through mowing, fire, and herbicide application (hereafter termed non-forested) or fire-maintained pine savanna with fuel beds including needle cast, turkey oak litter, herbaceous and shrub vegetation, and woody material (hereafter termed forested). Non-forested blocks included large (L1G and L2G) and small burn blocks (S3, S4, S5, S7, S8, and S9) while there was a single large forested block (L2F). Burn blocks, fuels, and fire behavior are described in and Butler et al. [19], Dickinson et al. [33], Hudak et al. [34], and Ottmar et al. [32] and are summarized in Table 1. Near-source plumes, including sensible heat fluxes, are described in Clements et al. [35].

### 2.2. Sensible Heat Flux and Energy

Sensible heat fluxes from the flow of hot gases are estimated from gas temperatures measured with fine thermocouples and velocities measured with temperature-compensated bi-directional probes and fast-response pressure transducers. We calculate the horizontal and vertical perturbation heat fluxes and energy and their resultant. In this context, perturbation refers to a departure from the pre-fire background state. Perturbation values are assumed from here forward. Accordingly, horizontal velocity and temperature are as follows:(2)$$u^{\prime }=u-\bar{u}$$
(3)$$T^{\prime }=T-\bar{T}$$
where *u* and *T* refer to the instantaneous values, and the overbar refers to the pre-fire average. Streamwise horizontal velocity is positive if its towards the front of the FBP and negative if from the rear. The horizontal sensible heat flux (*H’_u_*) is:(4)$$H_{u}^{\prime }=\rho C_{P}u^{\prime }T^{\prime }$$
where *ρ* is gas density (kg/m^3^), *C_P_* is specific heat capacity (J/kg K), *u′* is the streamwise velocity (m/s), and *T′* is the temperature (K). Gas density and heat capacity are temperature dependent (see Appendix A). Vertical (upward) velocity (analogous to Equation (2)) is positive while downward flow is negative. The vertical sensible heat flux is analogous to Equation (4). The FBPs were positioned so that the fire would generally approach from the front (defined by positioning of the incident radiant and total heat flux sensors).

Time-integration of horizontal and vertical sensible heat flux provides sensible energy (kJ/m^2^). The integration is limited by the residence time (*t_R_*), shown here for horizontal sensible energy (*E_Su_*):(5)$$E_{S_{u}}=t\overset{t_{R}}{\underset{1}{\sum }}H_{u}^{\prime }$$
where the time-step (*t*) is 0.1 s. Residence time is defined below. Vertical sensible energy is analogous to Equation (5).

We are also interested in the resultant sensible energy which we define as the resultant of the horizontal and vertical sensible energies. Preferable would have been to calculate the resultant of instantaneous horizontal and sensible heat fluxes [17], but the separation of horizontal and vertical probes may invalidate this approach. Instead, we estimate resultant sensible energy (*E_Sr_*) from horizontal and vertical energies as:(6)$$\;E_{S_{r}}={\left[{\left(E_{S_{u}}\right)}^{2}+{\left(E_{S_{w}}\right)}^{2}\right]}^{\frac{1}{2}}$$
where, again, density and heat capacity are temperature dependent and *w* refers to vertical.

We calculate perturbation energies, that is, we remove the contribution of pre-fire sensible heat fluxes and focus on the fire residence time, because we are interested in balancing energy generation from fuel combustion (left-hand side of Equation (1)) and energy sinks and dissipation mechanisms. Accordingly, we forego the use of Reynolds (moving) averaging through the residence time required to isolate turbulent kinetic energy from total sensible energy [17,37]. Instead, the perturbation sensible energies are estimates of fire totals. As described below, the constraints for estimating total sensible energy from a two-dimensional system of probes include the requirement that fires are spreading with the average flow in a direction in line with the horizontal flow measurement.

### 2.3. Instruments and Measurements

Estimating sensible heat fluxes and energy requires gas velocities and temperatures (Equations (2)–(4)). The core instruments are deployed in a Fire Behavior Package (FBPs) described in Butler et al. [38] and shown in Figure 1. The FBP includes one vertically- and one horizontally-oriented bi-directional pressure probe with +/−60 degree directional sensitivity [20] and a fine bare Type-K thermocouple (nominally 0.025 mm bead diameter) for temperature measurement. The probe characteristic dimension [20] is 12.7 mm. The ends of the fine thermocouple are welded to their corresponding leads and are not visible in Figure 1. The bi-directional probes are separated in space by approximately 36 cm while the thermocouple is positioned between the probes. The vertical probe connects to the FBP container on the left side (when the viewer faces the front of the FBP) and the horizontal probe joins the container on the top (Figure 1). Tubing transmits pressure signals from the dynamic and static sides of each probe separately to differential pressure sensors (Omega Engineering model PX137-0.3DV) which are temperature compensated with a pressure range of approximately ± 2000 Pa. Apart from instruments used to measure gas velocity and temperature, the FBP includes a Medtherm^®^ Dual Sensor Heat Flux sensor (Model 64-20T) that measures incident radiant and total (convective plus radiant) heat deposition onto the face of the sensor and a custom narrow angle radiometer (NAR, [39]) to characterize flame emissive power. The Dual Sensor and NAR are mounted flush with the FBP’s container and are oriented towards oncoming fires as best as can be predicted prior to ignition. Data from Dual Sensor and NAR are reported elsewhere [19]. Each FBP additionally contains a Campbell Scientific^®^ model CR1000 datalogger, two battery packs, and electronics required for each instrument. The sampling interval for all measurements is 10 Hz. The container is covered with two outer layers of fire-shelter material with an inner core of ceramic fiber insulation to prevent excessive heating. The FBP is elevated on a fire-hardened tripod to, nominally, 50 cm aboveground [19]. As such, approximately, the vertical probe is at 50 cm, the horizontal probe is at 81 cm, and the thermocouple is at 66 cm height aboveground.

Lacking local fuel consumption data, we infer fire energy (left-hand-side of Equation (1)) with data from overhead (nadir) radiometers [1,33]. We estimate fire energy from fire radiated energy density (FRED, also known as fire radiated energy [FRE]) and an assumption, based on measurements, that 17% of energy generated by combustion was radiated [33,34]. Two radiometer configurations were used, one elevated to 5.5 m on a tower with a 52 degree field of view and 22.5 m^2^ area of regard [33] and the second elevated to 7.7 m with a 60 degree field of view and 62 m^2^ area of regard [40]. Radiometer height does not affect the energy estimate [1] other than the area over which it is determined. FBPs were positioned just outside the area of regard of the radiometers and oriented across the area of regard and towards the expected approach of the flame front. 

Gas heat capacity and pressure (Equation (4)) and the calibration process required to estimate flow velocity are temperature dependent (Appendix A). We did not adjust for differences between air and flame and plume gases in their physiochemical properties. We used a lookup table to adjust air heat capacity by temperature [41]. Flow velocity, assuming incompressible flow, is derived by calibration from wind tunnel data (Appendix A). By convention, direction is determined by the sign of the differential pressure measurement with upward being positive on the vertically oriented bi-directional probe and flow towards the front of the Dual Sensor being positive on the horizontally oriented bi-directional probe (see Figure 1).

Time limits to the integrals used to determine sensible energy are determined using 3-s averaged gas temperature measurements because turbulence results in high-frequency fluctuation in temperatures and flows [37,42]. The start of the integral is determined by the timestep at which a 4-s window moving back in time from peak temperature last contains a temperature rise above threshold. We tested 50, 100, and 200 °C rise above background as thresholds. The end of the integral is the timestep at which a window moving forward in time from peak temperature encounters its last rise above threshold within the window. The limits to the integral define what we call residence time in this paper.

### 2.4. Statistics

Where correlations are reported, they are nonparametric Spearman rank-order correlation coefficients. Comparisons among thresholds used to define residence times are compared by ANOVA on log-transformed data. Within temperature thresholds, comparisons between horizontal and vertical flow velocities and sensible heat fluxes are by paired t-test. Regressions between horizontal, vertical, and resultant sensible energy and fire energy were linear on natural-log transformed data. Statistics were calculated with SAS 9. The standard for judging whether a difference between groups was significant was *p*
< 0.05.

## 3. Results

Vertical, horizontal, and resultant sensible heat fluxes and energies (Equations (5) and (6)) are based on flow velocity and direction and gas temperature measurements and inferred temperature-dependent gas density and heat capacity (Equations (2)–(4)). Individual collections were included in the final dataset if we knew from video analysis that the fire approached the FBP from within 60 degrees of perpendicular to the face of the incident heat flux sensor and axis of the horizontal probe (see Dickinson et al., 2019). Where we did not have video information, we further included datasets where sensible heat fluxes in the horizontal and vertical dimensions, and their resultant, were positive on average. We excluded datasets that did not meet the above conditions or which had known equipment problems. Ultimately, we report data from 55 out of 97 datasets.

### 3.1. Residence Times and Gas Temperatures

After experimenting with a range of thresholds, we ultimately used a 50 °C temperature rise above background to determine the residence times over which sensible heat fluxes were integrated (Figure 2). The choice was based on the objective of capturing as much of the perturbation (fire-induced) sensible heat flux and energy as possible. The temperature-rise rule provided a consistent estimate that accounted for variation in ambient (pre-fire) air temperature. We expect that there was minimal heat flux lost by excluding near-ambient temperatures. In practice, the estimates of horizontal, vertical, and resultant sensible energy did not differ among the 50, 100, and 200 °C thresholds (ANOVA F-value < 0.9, *p* > 0.4 for all comparisons) although the number of experiments that met the residence time criteria declined as the temperature threshold increased. Residence time used for integration should not be confused with flame residence times which were determined from video analysis and are shorter in duration (Table 1). The frequency distribution of residence times (Figure 2) follows from the wide range of gas temperature regimes in the flames and plumes (Figure 3 and Figure 4).

### 3.2. Flow Velocity and Horizontal and Vertical Sensible Heat Flux

The frequency distribution of average horizontal and vertical flow velocities showed a wide range across collections (Figure 5) and were not correlated which each other (R = −0.1, *p* = 0.6). Average horizontal velocities were larger than vertical velocities (Figure 5, Table 2). Peak flow velocities (Figure 6) show high instantaneous values, particularly for horizontal flow. Average sensible heat fluxes were greater in the horizontal than vertical directions (Table 2, Figure 7). As for flow velocities, horizontal and vertical sensible heat fluxes were not correlated (R = 0.24, *p* = 0.07). Peak horizontal and vertical sensible heat fluxes were often large (Figure 8) but fluctuated dramatically in our 10 Hz data (Figure 9). Residence times were strongly correlated with vertical sensible energy (R = 0.6, *p* > 0.0001) but were weakly correlated to horizontal sensible energy (R = 0.26, *p* = 0.06). The increase in residence time with sensible energy can be seen in Figure 9 in a comparison between timeseries with the lowest and median resultant sensible energies and that from the greatest resultant sensible energy.

### 3.3. Horizontal, Vertical, and Resultant Sensible Energy

Horizontal sensible energies, the time-integral of sensible heat fluxes (Figure 5 and Figure 6), were larger than vertical sensible energies (Table 2, Figure 10). Horizontal, vertical, and resultant sensible energies were positively related, with considerable variability, to fire energy inferred from fire radiated energy density (Table 3, Figure 11). The slopes in Table 3 are estimates of the fraction of fire energy dissipated by sensible energy (Equation (1)). Presence of flame, or the hot plume above flames, at the FBP is indicated by peak sensible heat fluxes in Figure 9. Clearly, residence times for integration (Figure 2) include fire-generated sensible heat flux from before and after flame arrival at the FBP.

## 4. Discussion

Horizontal, vertical, and resultant energies were estimated to be 75%, 54%, and 64%, respectively, of fire energy generated by the RxCADRE surface fires in longleaf pine savanna and open ranges (Table 3, Figure 11). Clearly, because of their magnitude, sensible heat flux and energy are critical measurements for understanding fire and plume dynamics and fire effects and for closing the fire heat budget (Equation (1)). We believe that these are the first measurements of near-source sensible heat fluxes and energy from wildland fires using in situ (in and near-flame) instruments. Measurements were based on flow estimated from differential pressures using bi-directional probes [20] and gas temperatures from fine thermocouples. Measurements of heat dissipation from fires (Equation (1)) have heretofore been limited to radiation measurements [1,2,3,5] and one measurement of energy transferred into the soil [2] which we infer was ~5% of fire energy. Fractional radiated energy from field experiments are variable but average somewhere between 14 and 17% of fire energy [5]. The relative proportion of sensible versus radiant energy should increase with the size of flames (e.g., from surface fires to crown fires) through increases in flame emissivity [43,44]. As well, radiated fraction has been shown to decrease with increases in fuel moisture [45]. We recognize that there is substantial variability associated with our estimates of fractional energy (Figure 11) which we attribute, in part, to the fact that we inferred fire energy from radiation measurements and an unrealistic assumption of a constant radiated fraction. In order to balance the integrated wildland fire heat budget (Equation (1)) given the physical complexity of the problem, it is likely that measurements should be coordinated with physics-based fire modeling [46,47,48] within a well-defined control volume. Constraining the control volume is important. For instance, consider that an estimate of sensible energy from in situ instruments positioned in flames will likely be a larger fraction of fire energy than an estimate derived from a measurement in the lower plume because of the progressive loss of heat by radiation.

We recognize limitations that should be addressed in future designs of instruments for measuring sensible heat fluxes in wildland fires. McCaffrey and Heskestad [20] describe a polynomial relationship between the calibration coefficient and Reynolds number (Re) that asymptotes at Re > 1000, a value substantially exceeded in our experiments (Re ~8000) where high flow velocity (13 m/s, Figure 6) is combined with temperatures used to define integration limits (50 °C + ambient). Given generally positive relationships between gas temperatures and flow velocities, we expect that this situation occurred infrequently and, given low temperatures, would have had a small effect on sensible heat fluxes. In contrast, moderate temperatures combined with high flow velocities would also have yielded Reynolds numbers in the asymptotic range (Re ~3000) which suggests that we may have underestimated flow velocities under these conditions and, thus, underestimated sensible heat fluxes. We discuss instrument design improvements below and note that a probe with a smaller characteristic dimension than ours (12.7 mm) would reduce Reynolds numbers and may be appropriate for wildland fire measurement.

Our gas temperature measurements are from fine, exposed bead thermocouples that we know are biased estimators of gas temperatures [49,50]. As radiation loss declines and heat gain by convection increases as thermocouple diameter declines, fine thermocouples are more faithful indicators of gas temperatures than thick thermocouples [49]. Our 0.025 mm diameter thermocouples were near the lower limit for practical use in the field with finer thermocouples being too delicate. Based on Figure 1 in Walker and Stocks [49], we estimate that the error at a peak temperature of 1350 °C would be about 11 °C for our thermocouples which, at a peak velocity of 13 m/s translates to a reduction in peak sensible heat flux of about 1%. The error is not large, but only applies to peak temperatures. Clements and Seto [17] measured maximum horizontal Reynolds averaged heat fluxes of around 120 kW/m^2^ at 1.9 m above ground using sonic anemometry during a surface fire, a value substantially lower than the peak sensible heat fluxes we measured (Figure 8 and Figure 9). We attribute the difference to the Reynolds averaging, the greater height of their measurements, and the thicker thermocouples that they used to measure gas temperatures (nominally 0.08 mm). Clements and Seto [17] report 30 s Reynolds averaged values which are intended to isolate turbulent kinetic energy and not perturbation energies that we report which only remove pre-fire sensible heat flux and are appropriate for balancing the time-integrated wildland fire heat budget (Equation (1)). The error in peak sensible heat flux resulting from their 0.08 mm diameter thermocouples was probably not the largest source of the difference with our study. Based on data in Walker and Stocks [49], error in peak sensible heat fluxes from thermocouple error in their study was on the order of 3%. Similar considerations are relevant to Clements et al. [16,35]. Gas temperature measurement is clearly a limitation for achieving accurate estimates of sensible heat fluxes and energy. Shielded-aspirated thermocouples, as opposed to bare thermocouples, maximize heat gain from convection and neutralize heat loss from radiation in steady-state conditions [51,52]. A limitation is that aspirated thermocouples average over a sampling volume and are, thus, not true estimators of instantaneous gas temperatures. A blending of measurement with fine thermocouples and thermocouple heat budget modeling [53] may offer the best means of increasing accuracy of instantaneous gas temperature measurement.

Due to the separation between horizontally- and vertically oriented bi-directional probes (Figure 1), more appropriate for large flames, we are unsure of the error in flow direction based on a resultant of the two instantaneous measurements and we have not reported them here. Proximity of multi-dimensional flow measurements will deliver data that will provide more accurate estimates of flow direction [54]. Currently, the thermocouple used in the FBP (Figure 1) is located between the two probes, a design that is not ideal, but, we assume, supports estimates of sensible energy that are approximately correct after horizontal and vertical sensible heat fluxes are integrated over residence times.

Bi-directional probes oriented vertically and horizontally clearly miss flow in the third, that is, crossways direction. As such, even though the probes have a wide acceptance angle [20], flux and energy are underestimated. We attempted to minimize this problem by only accepting fires for which we had evidence that average flows were upwards and towards the front of the FBP (Figure 1). We do not pretend that we eliminated the problem, however, given that we had no information on crosswise flow and recognize that the turbulent flow that characterizes flames and plumes is three dimensional [37]. The high variability in sensible heat fluxes shown in Figure 9 reflect this turbulence and the intricate structure of turbulent diffusion flames [55]. Three-dimensional measurement would clearly be beneficial. We point the reader to a probe assembly involving two precisely oriented bi-directional probes that provides three-dimensional flow except under specific conditions [56]. Adding a third (crossways) bi-directional probe may be preferable to avoid limiting conditions [54]. Clearly, a worthy challenge for fire science is to develop accurate methods for measuring three-dimensional flow in situ.

A measurement limitation that is not easily addressed is the fact that sensible heat flux and energy measurements from bi-directional probes have an uncertain footprint because they are sensing at least some heat advected with the flow from approaching flame fronts and, after flame front passage, from heat sources behind the flaming front. Moreover, flame fronts are not steady state and, at best, approach a quasi-steady condition [57]. As such, it is not possible to know accurately the ground area over which the sensible energy was generated. The indeterminant footprint is indicated by the spikes in sensible heat fluxes before and after peak fluxes (originating from flames and hot gasses in the near-source plume) with a particularly long tail of sensible heat fluxes from the high energy fire on the right-hand side of Figure 9. We can speculate that high-frequency infrared imaging, particularly if done in three dimensions [58] could better describe the spatiotemporal field of sensible heat fluxes and energy [9]. Until spatial measurement approaches are developed, increased replication of relatively local measurements is the only option for better characterizing sensible heat fluxes and energy on an aerial basis.

We assume that sensible energy is largely kinetic and that potential energy is negligible in a flame front wherein flame velocities resulting from buoyancy-driven flow are at a quasi-steady state. We speculate that this assumption would be least tenable during periods where flame velocities were accelerating or decelerating and heat flux rates were increasing or decreasing rapidly. It is not clear how we could address the effects of unsteady fire behavior without measurements of the spatiotemporal field of flow velocities and sensible heat fluxes using imaging methods [9,58].

Although smoldering combustion of duff in the longleaf pine savanna (L2F) contributed about 25% to fire energy [34], we assume that the bulk of sensible heat flux and energy in RxCADRE surface fires was from flaming combustion. Flame residence times were on the order of 10 s in these fires (Table 1) while our residence times for integration ranged up to nearly 100 s (Figure 2). In fires with substantial duff and downed woody fuel combustion, an assessment of their contribution to sensible heat flux may be possible. Where warranted, contributions from residual combustion might be assessed by first estimating the contribution from flames using a residence time capturing peak fluxes and, then, estimate contributions from residual combustion by extending the trailing limit of the residence time. Ward et al. [59] combined flow with chemical emissions measurements to characterize the mass balance of the primary combustion products which they used to partition fuel consumption into contributions from flaming and smoldering. Mass balance measurements may offer an important contribution to balancing the heat budget (Equation (1)).

*In situ* physical measurements of energy transport in flames and energy deposition in the wildland fire environment are critical for advancing fire science [18] yet are relatively sparse in the literature [42,58,60]. With the advent of modern numerical computation, the physical complexity and computational requirements of wildland fire behavior and effects models has increased, including models designed to simulate fire behavior [46,47,61,62,63], plume transport [64,65], and fire effects [66,67,68]. More physically realistic models and better basic understanding of fire dynamics require continued measurement development particularly of the basic heat and chemical processes occurring in fires exemplified by our study and others in the field [17,69,70,71,72,73,74] and laboratory [75,76,77,78,79,80]. In situ measurements also support remote sensing. Remote sensing measurements directly related to fire heat dissipation are currently limited to infrared radiation which can be quantified at high spatial and temporal resolution over large spatial extents [81]. Sensible heat fluxes and flow fields derived from coincident nadir radar and infrared measurements may, with more development, provide similar temporal and spatial coverage [9,82]. There is a need to add value to remotely-sensed measurements through in situ measurements that improve understanding of their physical connections with combustion and energy transport processes in fires [1,5,18] and, in turn, their links with plume transport [65] and fire effects [36].

We describe in situ measurements of sensible heat flux and energy in and near wildland surface fires. Closing the wildland fire heat budget (Equation (1), [1]) remains elusive yet is a scientific quest worth pursuing because doing so will help fire scientists better test their understanding of energy transport and heat deposition in the fire environment [19,58], will support smoke plume [65] and fire effects model development [25,26,27,36], and will improve remotely sensed measurements of fire behavior and energy [2,18]. Closing the fire heat budget requires continued measurement development and continued coordinated measurement campaigns over a wide range of fuel and fire behavior characteristics (e.g., [32,65]). Coordinated measurements should target all components of the heat budget, including the heat source [13,14,15,70], heat sinks, latent energy [16], soil heating [2], radiation [1,2,3,5], and sensible energy [16,17]. In addition, combining measurement campaigns with fire physics modeling is imperative [83] for both advancing the science and furthering the development of wildland fire management decision support systems.

## 5. Conclusions

Measurements in and above flames from in situ instruments confirm that sensible energies are the dominant mode of heat dissipation from flaming combustion in wildland surface fires (Equation (1)). Measurement challenges remain, including obtaining accurate gas temperatures and three-dimensional flow velocities across the varying spatiotemporal field of spreading fires. Continued development of sensible heat flux and energy measurements is warranted by its magnitude and its importance as a core mechanism driving fire spread, plume rise, and fire effects.

## Figures and Tables

**Figure 1 sensors-21-02135-f001:**
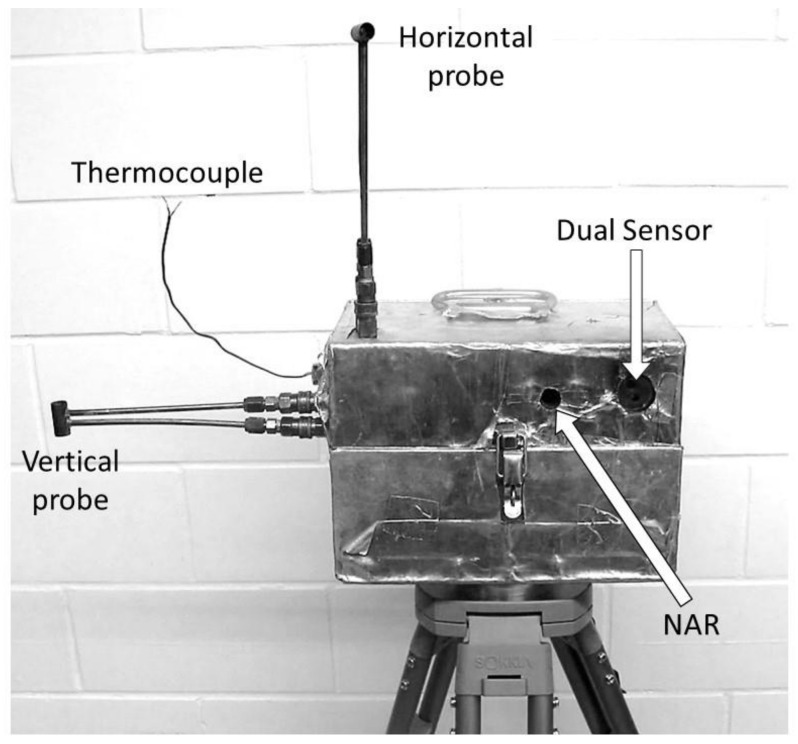
The Fire Behavior Package (FBP) including horizontally- and vertically oriented bi-directional probes and thermocouple. Data from the Dual Sensor and narrow-angle radiometer (NAR) are reported elsewhere [19]. Nominally, the FBP is positioned 50 cm aboveground and the distance between the probes is approximately 36 cm.

**Figure 2 sensors-21-02135-f002:**
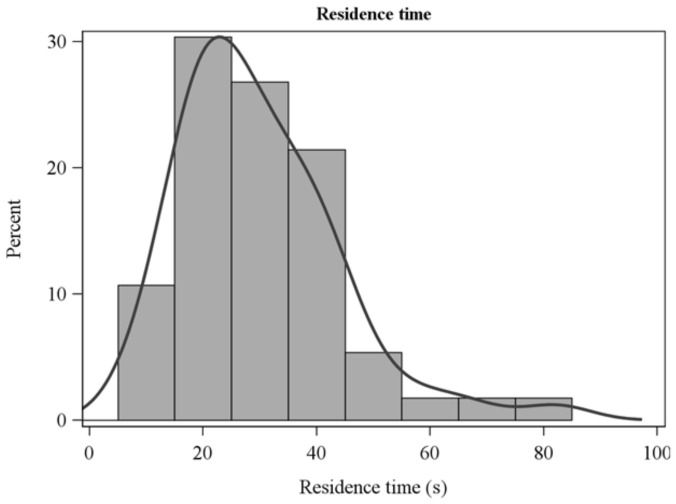
Frequency distribution of residence times defined by a 3-s moving-averaged temperature rise greater than 50 °C above background.

**Figure 3 sensors-21-02135-f003:**
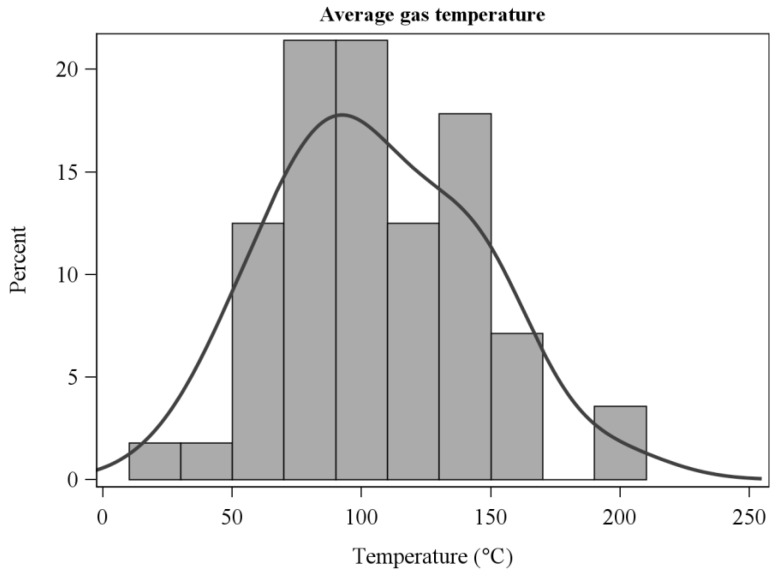
Frequency distribution of average gas temperatures over residence times measured with fine thermocouples.

**Figure 4 sensors-21-02135-f004:**
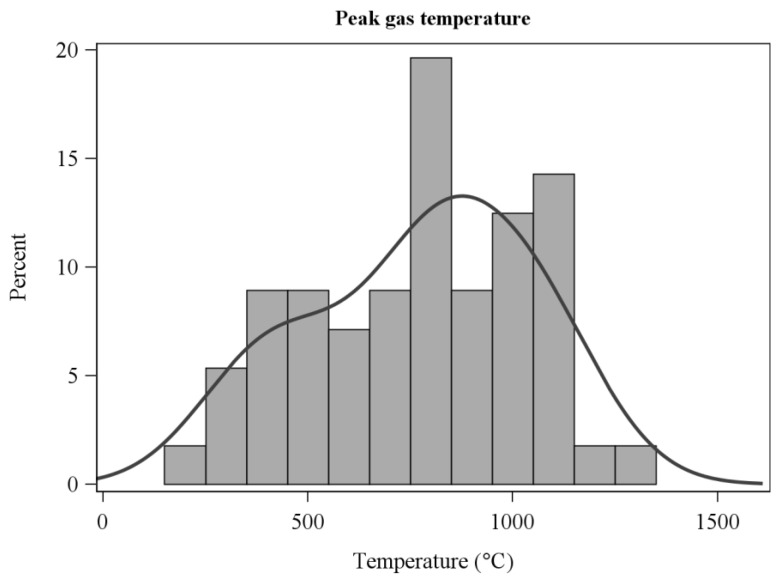
Frequency distribution of peak gas temperatures within residence times measured with fine thermocouples.

**Figure 5 sensors-21-02135-f005:**
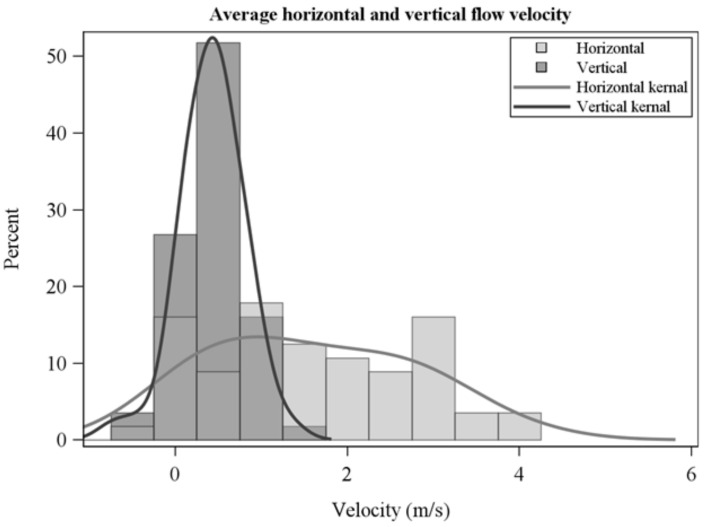
Horizontal and vertical velocities averaged over residence times.

**Figure 6 sensors-21-02135-f006:**
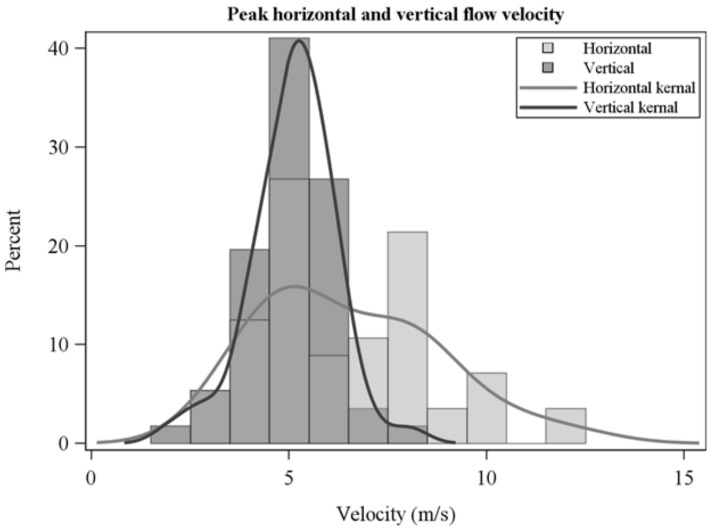
Peak horizontal and vertical flow velocities over residence times.

**Figure 7 sensors-21-02135-f007:**
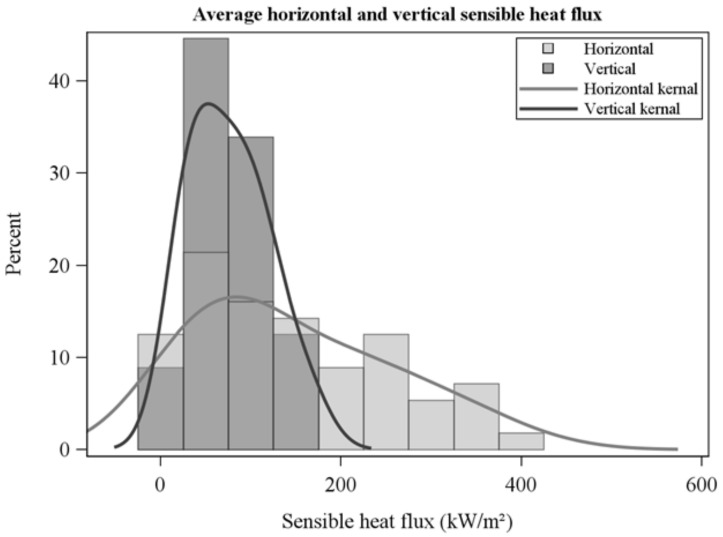
Average horizontal and vertical sensible heat fluxes over residence times.

**Figure 8 sensors-21-02135-f008:**
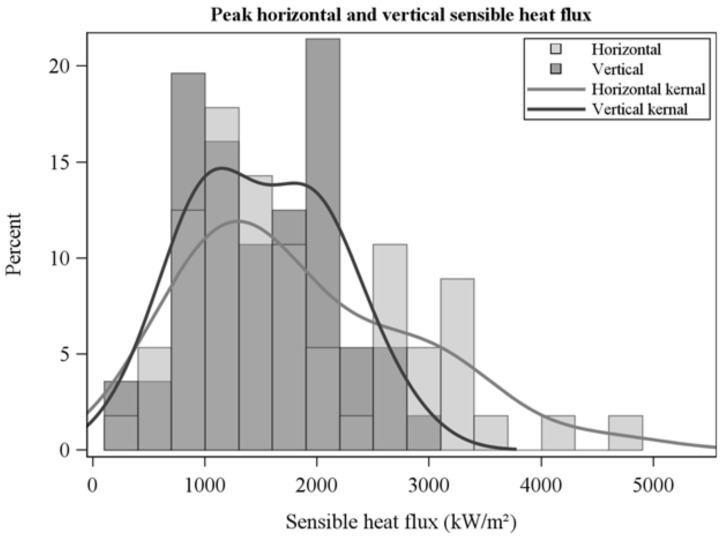
Peak horizontal and vertical sensible heat fluxes over residence times.

**Figure 9 sensors-21-02135-f009:**
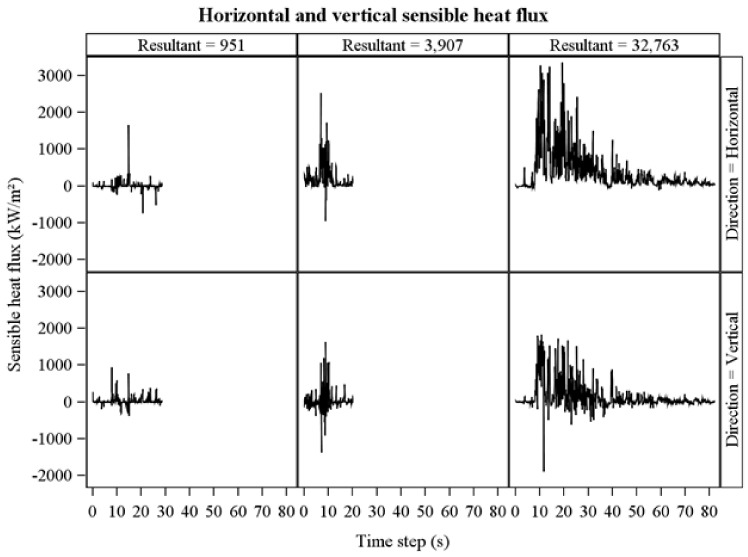
Time series of horizontal and vertical sensible heat flux (rows) across a range in resultant sensible energy (columns). Time series are from the datasets with the lowest (**left**), median (**center**), and highest (**right**) resultant sensible energies (kJ/m^2^).

**Figure 10 sensors-21-02135-f010:**
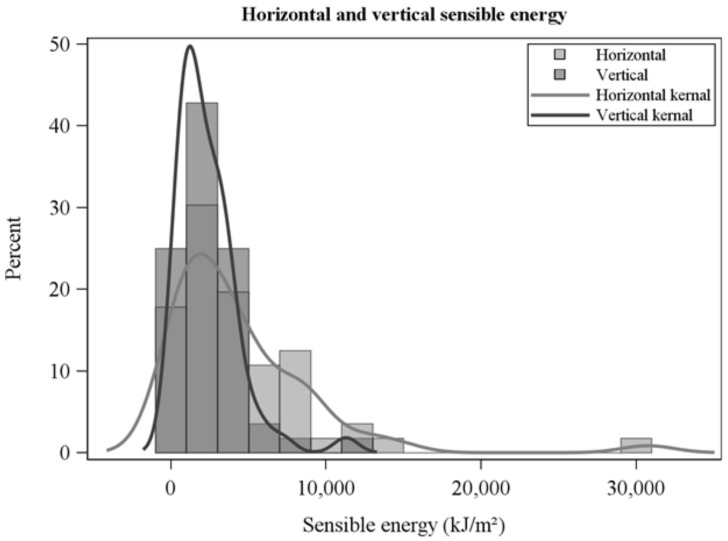
Horizontal and vertical sensible energies resulting from the time-integration of sensible heat fluxes over residence times.

**Figure 11 sensors-21-02135-f011:**
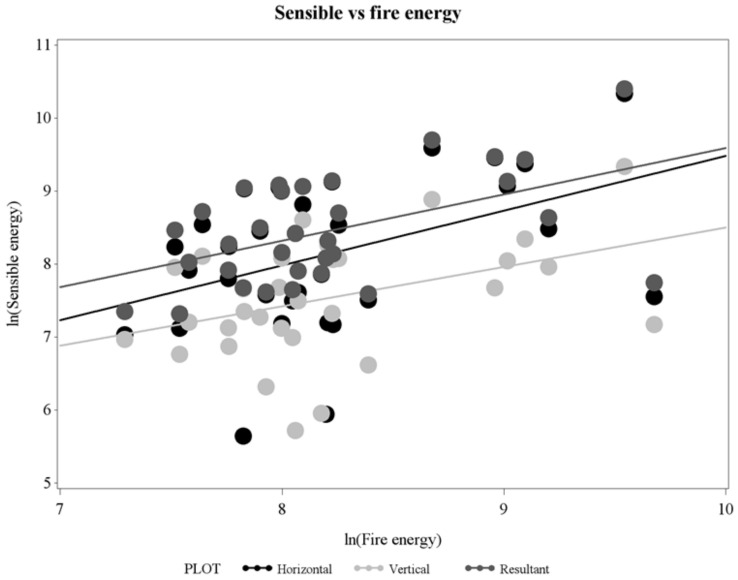
Horizontal, vertical, and resultant sensible energies (kJ/m^2^) as a function of fire energy (kJ/m^2^) estimated from overhead radiometers on natural log-transformed axes.

**Table 1 sensors-21-02135-t001:** Averaged characteristics of surface fires and flames in forested and non-forested burn blocks from the RxCADRE 2012 fires. Fireline intensity (*I*) and fuel consumption (*W*_1_) are inferred from nadir radiometer measurements using equations in Kremens et al. (2012, see Supplementary Material) while whole-block estimates of consumption (*W*_2_) are from Hudak et al. (2016). Estimates of flame height (*H_F_*), depth (*D_F_*), residence time (*t_R_*), and fire rate of spread (ROS) are from video analysis [19,36]. Sample sizes and standard deviations are provided (in parentheses) where applicable.

Burn block	Fuel	Date	*I* (kW/m)	*W*_1_ ((Mg/ha)	*W*_2_ ((Mg/ha)	*H_F_* ((m)	*D_F_* ((m)	*t_R_* ((s)	ROS ((m/s)
L2F	Forested	11/11/2012	907 (9, 670)	5.0 (9, 2.6)	6.4	0.9 (5, 0.5)	1.3 (5, 0.7)	9 (8, 7)	0.04 (2, 0.05)
L1G	Non-forested	11/04/2012	529 (9, 316)	1.3 (9, 0.5)	1.5	0.7 (6, 0.5)	1.1 (5, 0.8)	11 (7, 7)	0.24 (4, 0.30)
L2G	Non-forested	11/10/2012	739 (12, 358)	1.5 (12, 0.6)	3.1	0.5 (9, 0.2)	0.8 (9, 0.4)	11 (8, 6)	0.89 (3, 0.38)
S3	Non-forested	11/01/2012	479 (5, 79)	1.7 (5, 0.2)	2.6				
S4	Non-forested	11/01/2012	234 (4, 172)	1.6 (4, 0.7)	2.0				
S5	Non-forested	11/01/2012	564 (5, 269)	2.2 (5, 0.6)	2.2	0.4 (4, 0.0)	0.8 (4, 0.3)	11 (4, 4)	0.36 (2, 0.28)
S7	Non-forested	11/07/2012	1179 (4, 641)	3.3 (4, 1.8)	1.8				
S8	Non-forested	11/07/2012	512 (4, 318)	1.9 (4, 0.7)	2.8				
S9	Non-forested	11/07/2012	861 (5, 115)	1.8 (5, 0.9)	1.4				

**Table 2 sensors-21-02135-t002:** Comparisons between horizontal and vertical flow velocities and sensible heat fluxes averaged over residence times and sensible energies. Overall means and, in parentheses, standard deviations and ranges of average values along with t-test statistics are reported.

Dependent Variable	N	Horizontal Mean (SD, Range)	Vertical Mean (SD, Range)	*t*-Value	*p*
Flow velocity	55	1.6 (1.2, −0.4–4.1)	0.4 (0.4, −0.7–1.3)	7.1	<0.0001
Sensible heat flux	55	148 (110, 4–423)	76 (43, 11–171)	4.6	<0.0001
Sensible energy	55	4477 (4978, 171–30753)	2334 (1941, 208–11298)	3.0	0.003

**Table 3 sensors-21-02135-t003:** Linear regression statistics for horizontal, vertical, and resultant sensible energies as a function of fire energy (see Figure 11). Dependent and independent variables were log-transformed.

Dependent Variable	N	Intercept (ln[kJ/m^2^])	Slope (Dimensionless)	R^2^	*p*
Horizontal	32	2.00	0.75	0.18	0.015
Vertical	32	3.12	0.54	0.16	0.003
Resultant	32	3.25	0.64	0.26	0.003

## Data Availability

Radiometer and FBP data [84,85], and instrument locations [86,87], and other RxCADRE data are available on the USDA Forest Service Research Data Archive.

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
