# Peer review of "The Wildland Fire Heat Budget—Using Bi-Directional Probes to Measure Sensible Heat Flux and Energy in Surface Fires"

_sensors, 2021, doi:10.3390/s21062135_

Round 1

Reviewer 1 Report

The application of bi-directional probes to measure sensible heat flux and energy in the wildland fire is considerable challenging. As mentioned in the manuscript, the current measurement method requires additional improvement for accurate temperature and velocity measurements. Nevertheless, given the tremendous scale and danger in wildland fires, the applied measurement technology is a useful attempt to understand the fire phenomena. The purpose of the study, measurement method, and analysis of results were presented systematically, therefore, publication of this manuscript is recommended after the minor revision.

The use of bi-directional probes in wildland fire situations is quite efficient. However, it is known that the probe coefficient can vary significantly depending on the Re number and a fixed probe coefficient can be used only under conditions of sufficient turbulent flow. Therefore, more specific verification and limitations on the method of calculating the velocity using the bi-directional probe should be presented.

Also, considering the complex three-dimensional flow field near the ground, it is questionable whether the attacking angle of the bi-directional probe (e.g. 60 degree) is acceptable for wildland fires.

A bare-bead thermocouple was used for temperature measurement, but the measurement error due to radiant heat in the wildland fire is expected to be very large. Further explanation should be presented for this.

The position of the thermocouple was selected in the middle of the horizontal and vertical bi-directional probe. In wildland fire, if there is a significant difference in gas temperature depending on the altitude near the ground, this can also cause a large error in the prediction of the sensible heat flux. A description of this should be provided in the revised manuscript.

Reviewer 2 Report

This article proposes a data analysis of measurements performed with a special in site apparatus able to deduced sensible heat flux and energy from the measurement of temperature and velocity.

The paper is clear, well-presented and well-written. The references are appropriate. The results clearly demonstrate the performance of such new apparatus. But it also points out the limits and the perspectives of improvement. The paper is appropriate for publications.

Here are some questions and comments that may improve the paper :

  1. Appendice A: Usually a probe coefficient is included in relation 1 that take into account the special shape of the probe (in comparison to pitot probe shape). McCaffrey suggested a correlation for this coefficient. Why this coefficient is not introduced here ? Could you mention that point?
  2. Reference: They are also contributions using PIV and laser technique for assessing convective flows in wildland fire. You can also mention them.
